# Identification of EOMES-expressing spermatogonial stem cells and their regulation by PLZF

Manju Sharma, Anuj Srivastava, Heather E Fairfield, David Bergstrom, William F Flynn, Robert E Braun*

The Jackson Laboratory, Bar Harbor, United States

**Abstract** Long-term maintenance of spermatogenesis in mammals is supported by GDNF, an essential growth factor required for spermatogonial stem cell (SSC) self-renewal. Exploiting a transgenic GDNF overexpression model, which expands and normalizes the pool of undifferentiated spermatogonia between $Plzf^{+/+}$ and $Plzf^{lu/lu}$ mice, we used RNAseq to identify a rare subpopulation of cells that express EOMES, a T-box transcription factor. Lineage tracing and busulfan challenge show that these are SSCs that contribute to steady state spermatogenesis as well as regeneration following chemical injury. EOMES+ SSCs have a lower proliferation index in wild-type than in $Plzf^{lu/lu}$ mice, suggesting that PLZF regulates their proliferative activity and that EOMES+ SSCs are lost through proliferative exhaustion in $Plzf^{lu/lu}$ mice. Single cell RNA sequencing of EOMES+ cells from $Plzf^{+/+}$ and $Plzf^{lu/lu}$ mice support the conclusion that SSCs are hierarchical yet heterogeneous.

DOI: https://doi.org/10.7554/eLife.43352.001

## Introduction

Fertility in males is supported by a robust stem cell system that allows for continuous sperm production throughout the reproductive life of the individual. In humans this lasts for decades and in a mouse can last for nearly its entire lifetime. However, despite more than a half century of research, and intensive investigation by many labs over the last decade, the identity of the germline stem cell continues to be elusive and controversial.

Stem cell function clearly resides in a subpopulation of spermatogonia within the basal compartment of the seminiferous tubules. In 1971, Huckins and Oakberg proposed the 'As' model of spermatogonial stem cells (SSC) function where $A_{single}$ ($A_s$) SSCs divide in a linear and non-reversible manner to populate the spermatogenic lineage (*Oakberg, 1971a*; *Oakberg, 1971b*; *Huckins, 1971*). In this model, $A_s$ spermatogonia are the SSCs, dividing symmetrically and with complete cytokinesis to form two daughter $A_s$ cells for self-renewal, or dividing with incomplete cytokinesis to form $A_{paired}$ ($A_{pr}$) cells, which are irreversibly committed to differentiation. $A_{pr}$ cells, in turn, divide to form $A_{aligned}$ ($A_{al}$) spermatogonia, which exist as chains of 4, 8, or 16 interconnected cells. The $A_s$, $A_{pr}$, and $A_{al}$ spermatogonia encompass the pool of undifferentiated spermatogonia that can be identified morphologically and are thought to share important functional properties distinct from the differentiated spermatogonia; $A_1$-$A_4$, intermediate and B (*Chiarini-Garcia and Russell, 2001*; *Chiarini-Garcia and Russell, 2002*).

Contrary to the Huckins/Oakberg $A_s$ model, recent studies on the behavior of pulse-labeled spermatogonia populations suggest that $A_{pr}$ and $A_{al}$ syncytia can fragment and revert to become $A_s$ cells following transplantation as well as during steady state spermatogenesis (*Nakagawa et al., 2010*; *Hara et al., 2014*). This suggests that undifferentiated spermatogonia are not irreversibly committed to differentiation, allowing for an alternative mechanism for SSC self-renewal. Furthermore, $A_s$, $A_{pr}$,

*For correspondence:
bob.braun@jax.org

Competing interests: The authors declare that no competing interests exist.

and $A_{al}$ spermatogonia are characterized by heterogeneous gene expression (*Nakagawa et al., 2010*; *Hara et al., 2014*; *Suzuki et al., 2009*; *Nakagawa et al., 2007*; *Barroca et al., 2009*; *Sada et al., 2009*) including recent descriptions of specific subpopulations of $A_s$ cells with SSC activity expressing *Id4, Pax7, Bmi1, T* and *Pdx1* (*Chan et al., 2014*; *Aloisio et al., 2014*; *Komai et al., 2014*; *Tokue et al., 2017*; *La et al., 2018*). These new data do not easily comport to a unifying model and imply that the mode of SSC function in the testes is more complex than the original Huckins-Oakberg $A_s$ model suggests.

A majority of $A_s$ and $A_{pr}$ cells express GFRA1, a GPI-anchored receptor for glial cell-derived neurotrophic factor (GDNF) (*Buageaw et al., 2005*; *Naughton et al., 2006*; *Johnston et al., 2011*; *Sato et al., 2011*; *Grasso et al., 2012*). GDNF is secreted by neighboring somatic Sertoli (*Meng et al., 2000*) and peritubular myoid (*Chen et al., 2016*) cells and is required for establishment and self-renewal of the SSC population in a dose-dependent manner (*Meng et al., 2000*). A decrease in GDNF levels results in germ cell loss, while overexpression of GDNF promotes accumulation of SSCs due to a block in differentiation (*Meng et al., 2000*; *Sharma and Braun, 2018*). *Plzf* (*Zbtb16*), which encodes a transcription factor expressed in $A_s$, $A_{pr}$ and $A_{al}$ spermatogonia, is required for SSC maintenance, as mutation of *Plzf* results in age-dependent germ cell loss (*Buaas et al., 2004*; *Costoya et al., 2004*). The mechanisms by which PLZF regulates SSC maintenance are not yet known. We describe here the identification of a rare subpopulation of $A_s$ cells whose cycling frequency is altered in *Plzf* $^{lu/lu}$ mutants, suggesting that PLZF regulates the proliferation of SSCs.

## Results

### GDNF increases the undifferentiated spermatogonial population in *Plzf* mutants

Stage-specific temporal ectopic expression of GDNF in supporting Sertoli cells results in the accumulation of large clusters of tightly-packed PLZF+ undifferentiated spermatogonia (*Sharma and Braun, 2018*; *Yomogida et al., 2003*). To determine whether overexpression of GDNF could rescue germ cell loss in *luxoid* (*lu*) mutants, we generated Tg(*Ctsl-Gdnf*)$^{1Reb}$; *Plzf* $^{lu/lu}$ mice (referred to as Tg (*Gdnf*);*lu/lu*). While no difference was found in the testis/body weight ratio between WT and Tg (*Gdnf*) mice, the testis/body weight ratio was significantly higher in Tg(*Gdnf*);*lu/lu* mice compared to *lu/lu* (p=0.0005), although it was still lower than in Tg(*Gdnf*) (p=0.0001) animals (*Figure 1A*). At four months of age, Periodic acid-Schiff staining of testes sections showed fewer agametic tubules (referred to as Sertoli cell only) in Tg(*Gdnf*);*lu/lu* mice compared to *lu/lu* at both 4 and 6 months of age (*Figure 1B and C*). Increased testis/body weight in Tg(*GDNF*);*lu/lu* mice could therefore be due to an increase in the number of cells occupying individual tubules, reflected by a decrease in the number of tubules with a Sertoli cell only phenotype, and fewer tubules with loss of one or more cell populations at 6 months of age (*Figure 1C*).

To determine what germ cell populations were expanded in Tg(*Gdnf*);*lu/lu* testes, we immunostained both whole-mount tubules and sections for spermatogonia markers. Large clusters of GFRA1 + cells were observed in Tg(*Gdnf*);*lu/lu* tubules, suggesting an increased number of the earliest spermatogonial cell types, $A_s$ and $A_{pr}$ (*Figure 1—figure supplement 1A*). Quantitation of LIN28A staining, which is detected in a subpopulation of $A_s$, and $A_{pr}$, $A_{al}$ and A1 spermatogonia (*Chakraborty et al., 2014*), on 12 week old testes sections (*Figure 1—figure supplement 1B*) showed that 75% of Tg(*Gdnf*);*lu/lu* tubule cross-sections had >5 LIN28A+ cells per tubule compared to 30% in *lu/lu* (*Figure 1—figure supplement 1C*, p<0.001). Only 8% of tubules lacked LIN28A-labeled cells compared to 30% in *lu/lu* testes (p<0.01). To determine whether the $A_1$ to $A_4$ differentiated spermatogonia population increased in Tg(*Gdnf*);*lu/lu* tubules, we immunostained for the marker SOHLH1, which is not expressed in PLZF+ $A_s$ to $A_{al}$ cells (*Figure 1—figure supplement 1D*). Similar to LIN28A, quantification of sectioned tubules revealed a significant increase in the percent of tubules with >5 SOHLH1+ cells (*Figure 1—figure supplement 1E,F*).

Using quantitative reverse transcriptase PCR (qRT-PCR), we compared mRNA levels of several known genes expressed in undifferentiated spermatogonia, using total RNA from 4 month-old *lu/lu* and Tg(*Gdnf*);*lu/lu* testes. Similar to our immunostaining results, we observed a significant increase in the undifferentiated spermatogonia markers *Gfra1, Lin28a* and *Sohlh1* in Tg(*Gdnf*);*lu/lu* compared

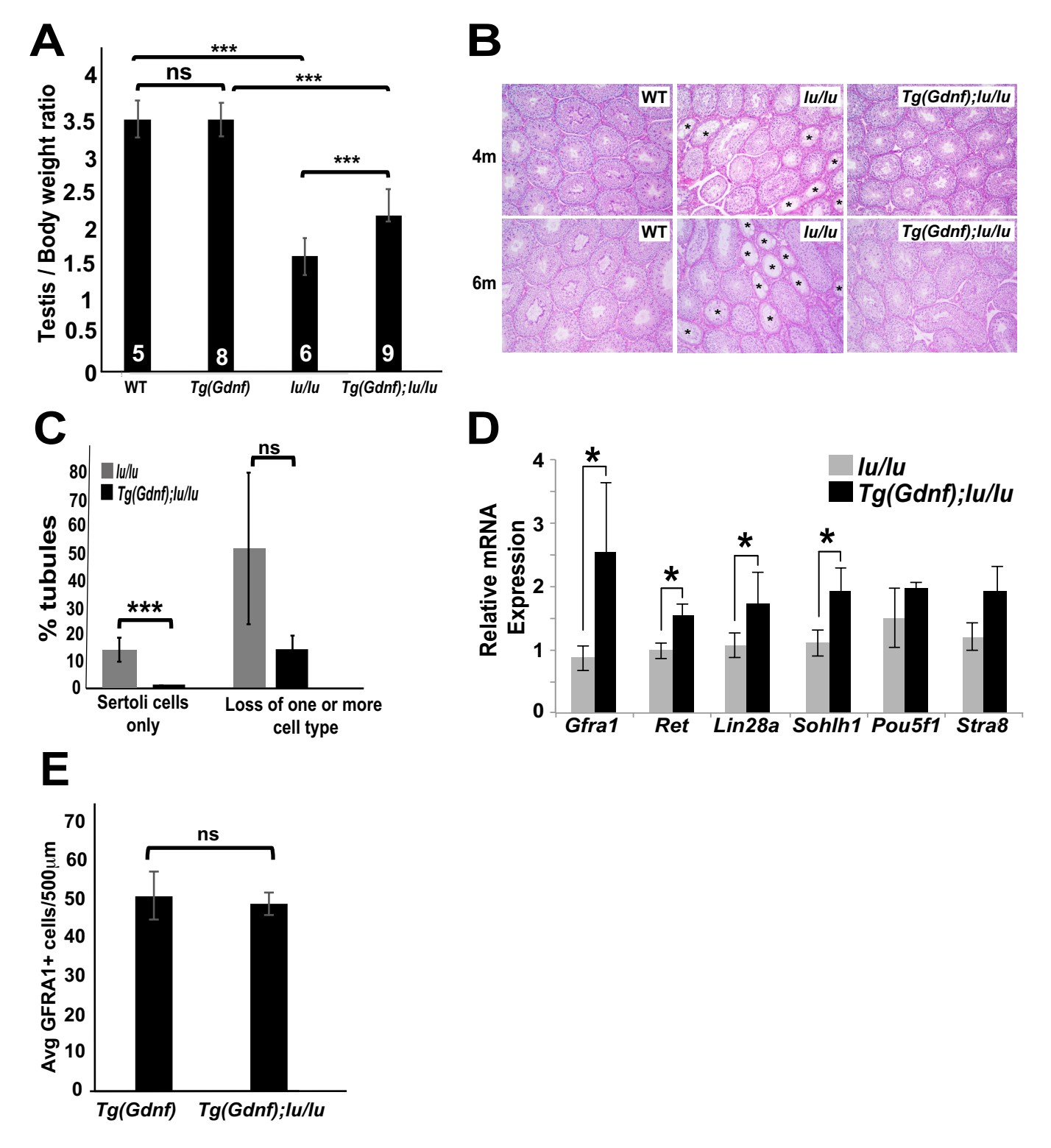

**Figure 1.** Sertoli-cell overexpression of GDNF partially reverses the loss of undifferentiated spermatogonia in *luxoid* mice. (**A**) Testis/body weight ratios. No significant (ns) difference was observed in testis/body weight between WT and Tg(*Gdnf*). Significant differences (***) were detected between WT and *lu/lu* (p=0.0001), Tg(*Gdnf*) and Tg(*Gdnf*);*lu/lu* (p=0.0001) and between *lu/lu* and Tg(*Gdnf*);*lu/lu* (p=0.0005). Number in bar = n animals. (**B**) Representative images of periodic-acid-Schiff stained cross-sections of 4- and 6 month old testes showing agametic tubules (asterisks) in *lu/lu*. (**C**) Fewer tubules with germ cell loss were present at 6 months in Tg(*Gdnf*):*lu/lu*. A total of 200–300 tubules were counted for each genotype (n = 3).
*Figure 1 continued on next page*

Figure 1 continued

p<0.005. (D) qRT-PCR on testis RNA from 4 month old mice shows a significant increase in SSC gene expression in *lu/lu* mice overexpressing GDNF in Sertoli cells. Relative mRNA levels are normalized to a *β-actin* internal control. *, p<0.01. (n = 3) (E) Average number of GFRA1+ cells per 500 μm length of tubules in Tg(*Gdnf*) and Tg(*Gdnf*);*lu/lu* in 10wk old mice (n = 3).

DOI: https://doi.org/10.7554/eLife.43352.002

The following source data and figure supplements are available for figure 1:

**Source data 1.** Source file for *Figure 1A, 1C and 1E*.

DOI: https://doi.org/10.7554/eLife.43352.005

**Figure supplement 1.** Sertoli-cell overexpression of GDNF increases the SSC population in *Plzf*-mutant *luxoid* mice.

DOI: https://doi.org/10.7554/eLife.43352.003

**Figure supplement 1—source data 1.** Source file for *Figure 1—figure supplement 1C and 1F*.

DOI: https://doi.org/10.7554/eLife.43352.004

to *lu/lu* (*Figure 1D*). We also saw an increase in *Ret*, which encodes the GFRA1 co-receptor. Thus, temporal ectopic expression of GDNF significantly increases the numbers of tubules with both undifferentiated and differentiated spermatogonia, leading to a partial rescue of testis weight in *lu/lu* mice.

## Unbiased discovery of genes whose expression is altered in *Plzf* mutants

Seminiferous tubules from Tg(*Gdnf*) mice have an expanded GFRA1+ population and enhanced ability to repopulate the spermatogonial lineage in testes transplants compared to WT (*Sharma and Braun, 2018*). Because Tg(*Gdnf*) and Tg(*Gdnf*);*lu/lu* testes have a comparable number of GFRA1+ spermatogonia (*Figure 1E*), we reasoned that these mice could be used as a valuable tool for the discovery of genes involved in self-renewal. Using RNAseq, we compared the transcriptomes of whole testes from Tg(*Gdnf*) and Tg(*Gdnf*);*lu/lu* mice and searched for genes with significant fold-changes in expression. Of 20,000 gene transcripts,~1500 were differentially expressed with an absolute $\log_2$ fold change >1.5 (FDR < 0.05). Of this subset, 150 were upregulated two-fold or more, and 60 were down-regulated two- to seven-fold in *luxoid* mutants.

As expected, many previously known gene transcripts specific to undifferentiated spermatogonia had no significant change in expression between genotypes including: *Gfra1, Ret, Lin28a, Utf1, Nanos2, Nanos3, Sohlh2,* and *Sall4* (*Figure 2A*), confirming that the undifferentiated spermatogonial populations were similar between the two genotypes. In the group of genes that were significantly down-regulated in Tg(*Gdnf*);*lu/lu* testes, we identified a number of genes that are involved in the maintenance of pluripotent stem cell fate or early embryonic development (*Figure 2B*) including: *Pou5f1, Brachyury (T)* and *Eomes*. *Pou5f1*, essential for maintaining pluripotency of ES cells and for maintaining viability of the mammalian germline (*Nichols et al., 1998*; *Kehler et al., 2004*), was down-regulated two-fold in *luxoid* testes overexpressing GDNF. *Brachyury (T)*, down-regulated seven-fold in Tg(*Gdnf*);*lu/lu* testes, is a T-box transcription factor important for germ cell development that when overexpressed results in testicular germ-cell tumors (*Sangoi et al., 2011*; *Tirabosco et al., 2008*). *Eomes*, a T-box transcription factor that was down-regulated 3-fold in Tg(*Gdnf*);*lu/lu* testes, is known to play a critical role in embryonic development (*Costello et al., 2011*; *Arnold et al., 2008*). Notably, the read counts per million (RCPM) for the two highly down-regulated genes shown, *Eomes* and *T*, were very low in either genotype and found at the distant end of the expression spectrum amongst 20,000 transcripts (*Figure 2C*, far right). *Eomes* had a value of only 1.4 RCPM in Tg(*Gdnf*) testis RNA compared to 5,509 RCPM for *Prm2*, a gene expressed in spermatids. This suggests that *Eomes* and *T* are targets of PLZF and are down-regulated in *lu/lu* mice, or that T and EOMES are expressed in a small subset of germ cells that are specifically reduced in *lu/lu* mutants, possibly within a restricted population of A$_s$ cells.

We next validated candidate genes at different developmental time points by RT-PCR. Using total testis RNA from Tg(*Gdnf*) and Tg(*Gdnf*);*lu/lu* mice, *Eomes* was reduced in *lu/lu* testes as early as 3 days after birth, and became barely detectable at 4 and 6 months of age (*Figure 2D*). As mutations in *Plzf* lead to age-dependent germ cell loss, EOMES is a compelling candidate for a marker of a functional pool of SSCs that is uniquely depleted in *luxoid* mutants.

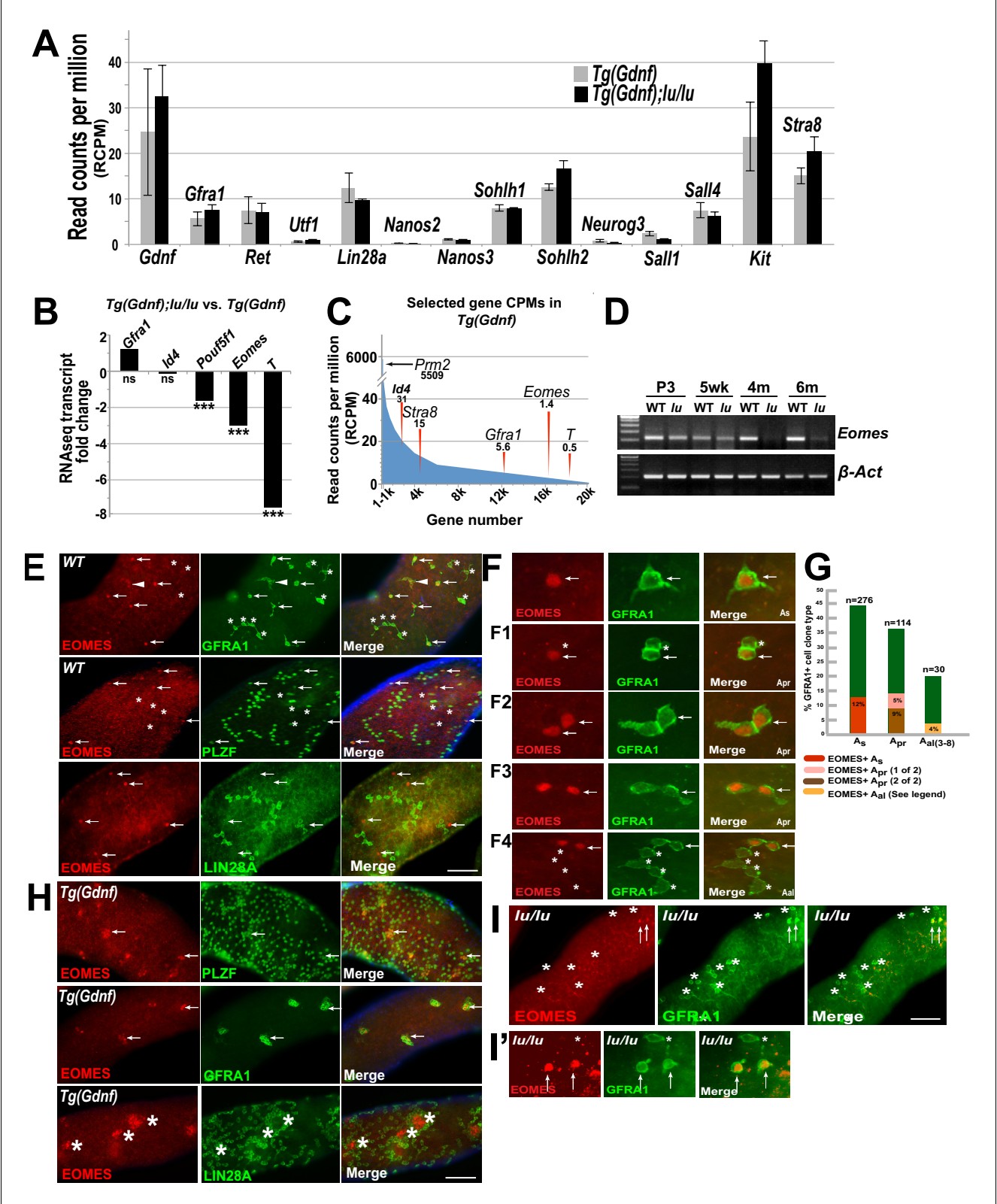

**Figure 2.** EOMES marks a subpopulation of SSCs. (A) Relative gene expression (read counts per million, RCPM) of selected transcripts from RNAseq performed on testis RNA from 4 month old Tg(*Gdnf*) and Tg(*Gdnf*);*lu/lu* mice. No significant change is detected in known spermatogonial marker genes. (B) Fold change in RNAseq transcripts for a subset of pluripotency and developmental genes in Tg(*Gdnf*);*lu/lu* compared to Tg(*Gdnf*) testes. ***, p<0.0001. (C) Relatively low RCPM of the highest-fold change transcripts shown in (B), compared to the spermatid-expressed protamine two gene

*Figure 2 continued on next page*

*Figure 2 continued*

*Prm2*, and differentiating spermatogonia marker *Stra8*. X-axis represents 20,000 genes numbered in descending order by RCPM. (D) Semi-quantitative RT-PCR of total testis RNA showing decreasing *Eomes* expression over time in *lu/lu* testes. (E) Immunostaining for EOMES and undifferentiated spermatogonial markers in WT whole-mount testes. EOMES marks some (arrows), but not all (asterisks), GFRA1+ PLZF+ $A_s$ cells. Weak EOMES staining can occasionally be detected in chains of GFRA1+ $A_{al}$ cells (arrowhead). EOMES+ $A_s$ cells do not express LIN28A (bottom row, arrows). (F) Examples of EOMES+ subset of GFRA1+ cells. F1 shows an example of two adjacent cells where one is EOMES+ (arrow) and one is EOMES- (asterisk). F2 and F2 illustrate two examples of two juxtaposed cells that are both EOMES+. F4 illustrates an example of a chain of 6 $A_{al}$ cells with two EOMES+ (arrow) and four EOMES– (asterisks) cells. (G) Graph showing the percent of EOMES+ cell in the GFRA1+ population clone types in WT adult mice (631 GFRA1 + cells from three mice). n = number of clone types counted (e.g. $A_{pr}$ = 114 pairs of cells = 228 GFRA1+ total cells). Of the total GFRA1+ $A_{pr}$ population, in 9% of cases both cells were positive for EOMES, while 5% had only one cell positive for EOMES. Within the $A_{al}$ fraction, 4% of the clones express EOMES in a heterogeneous pattern (two $A_{al}$ chains where 3 of 3 were EOMES+; two $A_{al}$ chains of 4 where all were EOMES+; one $A_{al}$ chain of 4 where only one was EOMES+; one $A_{al}$ chain of 6 where only 2 of 6 cells were EOMES+). (H) EOMES is expressed in tightly packed GFRA1+ PLZF+ clusters (arrows) found in Tg(*Gdnf*) whole-mount tubules. EOMES is not expressed in the LIN28A+ cells in the cortical region of the clusters (bottom row, asterisks). (I) Detection of EOMES+ GFRA1+ cells (arrows) in *Plzf* $^{lu/u}$ mutants. Asterisks mark an EOMES- GFRA1+ cell.

DOI: https://doi.org/10.7554/eLife.43352.006

The following source data is available for figure 2:

**Source data 1.** Source file for *Figure 2G*.

DOI: https://doi.org/10.7554/eLife.43352.007

## EOMES is detected in a subpopulation of GFRA1+ cells in the testis

Given the differential expression of *Eomes* in Tg(*Gdnf*) and Tg(*Gdnf*);*Plzf* $^{lu/lu}$ mutants, we assessed whether EOMES was expressed within the spermatogonial population of the testis. We immuno-stained whole mount WT seminiferous tubules for EOMES together with GFRA1, which marks $A_s$, $A_{pr}$ and small chains of $A_{al}$ spermatogonia (*Figure 2E,F*, arrows). EOMES was expressed in 12% of the total GFRA1+ $A_s$ population of cells (*Figure 2G*). Fourteen percent of GFRA1+ $A_{pr}$ clones were also EOMES+ in either one (5%) or both (9%) of the cells (*Figures 2F1, F and F3*, arrows, and G). The clones counted as $A_{pr}$ could be two lineage-independent, but adjacent, $A_s$ cells, $A_{pr}$ cells about to complete cytokinesis to form two new $A_s$ cells, or bona fide $A_{pr}$ cells that will go on to form a chain of $A_{al}$ cells. In rare cases (4%) EOMES was detected in some or all of the individual cells of GFRA1+ chains of $A_{al}$ cells (*Figure 2F4, G* and figure legend). In all instances, EOMES was co-expressed with PLZF (*Figure 2E*, arrows), which marks all undifferentiated spermatogonia. EOMES was not detected in LIN28A-expressing cells (*Figure 2E*), which we detect in a subset of $A_s$ cells and in most $A_{pr}$ and $A_{al}$ cells (*Chakraborty et al., 2014*).

Because the EOMES+ population also expressed GFRA1, we hypothesized that these cells might be under GDNF influence (*Meng et al., 2000*). We therefore asked whether Sertoli-specific overexpression of GDNF affected this cell population. In whole-mount Tg(*Gdnf*) tubules we detected EOMES+ cells in clusters of PLZF+ and GFRA1+ cells (*Figure 2H*, arrows). LIN28A stained only the peripheries of these cell clusters, and was specifically excluded from the EOMES+ cores (*Figure 2H*, bottom row, asterisks). These data indicate that EOMES primarily marks a subpopulation of cells that are GFRA1+, PLZF+ and LIN28A-.

To determine if EOMES-expressing cells are present in *lu/lu* testes, we immuno-stained *lu/lu* tubules for EOMES and GFRA1. Despite the significant decrease in *Eomes* RNA in whole testes, EOMES+ cells were detected in *lu/lu* mutants, (*Figure 2I and I'*, arrows), suggesting that PLZF regulates the pool of EOMES+ cells, but does not directly regulate EOMES expression.

## EOMES+ cells contribute to steady-state spermatogenesis

One of the features of stem cells is their ability to maintain their population throughout adult life while giving rise to differentiating progeny. To study the stem cell properties of EOMES expressing cells, we generated an inducible *iCreERt2/2A/tdTomato* knock-in allele (*Eomes* $^{iCreERt2}$) at the *Eomes* locus to label and trace their progeny after tamoxifen treatment (*Figure 3A*). Characterizing the mouse line for tdTOMATO expression, we observed single GFRA1+ tdTOMATO+ cells in whole mount seminiferous tubules from 4 week old males (*Figure 3B* top row arrows). EOMES immunofluorescence marked the same cells as tdTOMATO (*Figure 3B'*, top row), and as with EOMES itself, tdTOMATO was detected in a subpopulation of GFRA1+ cells (*Figure 3B'* bottom row, arrow). In testis sections, GFRA1+ tdTOMATO+ cells were detected within the seminiferous tubules

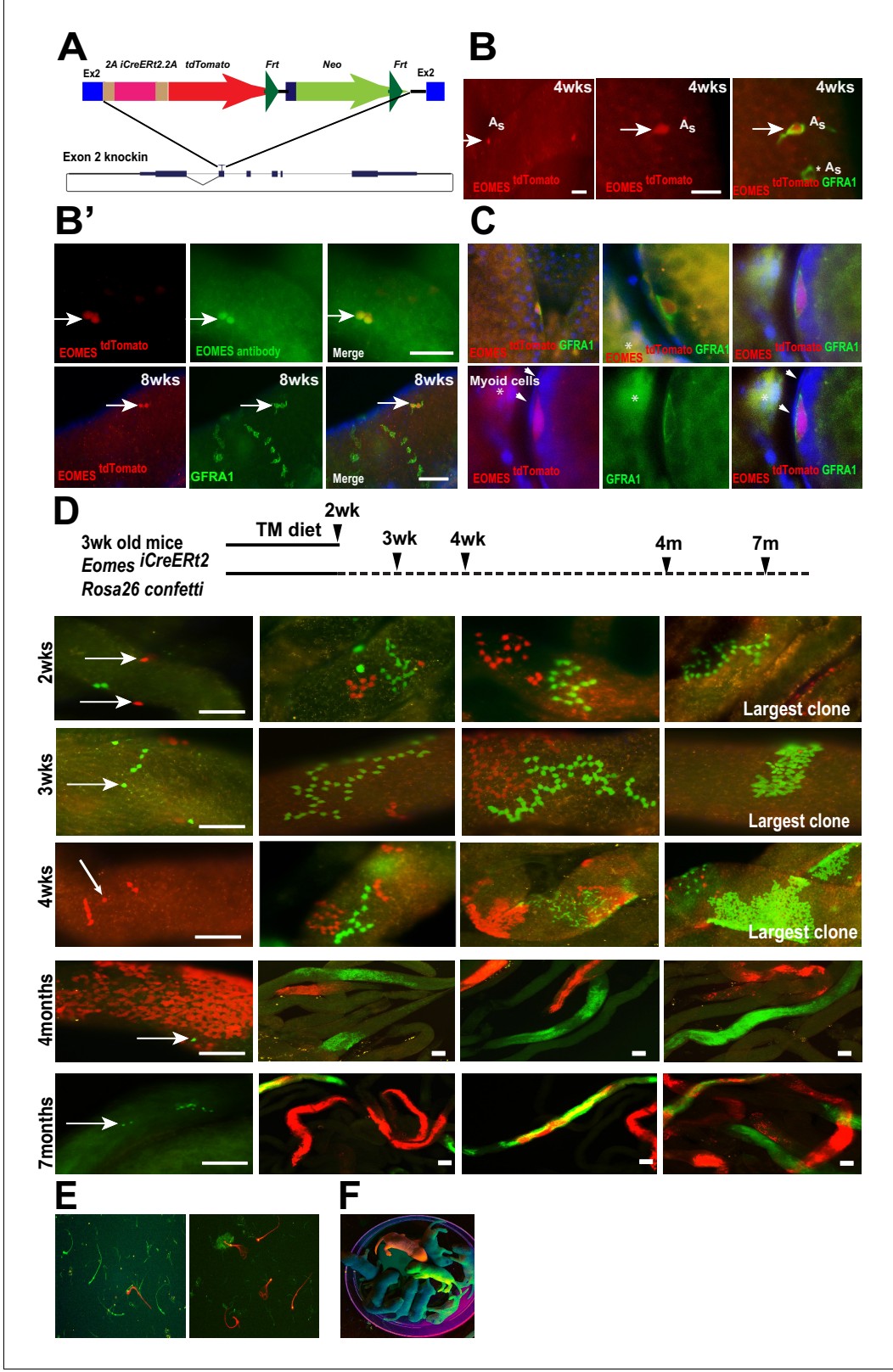

**Figure 3.** EOMES+ cells are long-term SSCs. (**A**) Schematic diagram of the *Eomes* locus on chromosome nine and insertion cassette used to generate an *Eomes/2AiCreERT2/2A/tdTomato* knock-in allele. (**B**) GFRA1+ tdTomato+ A_s cell in testis of 4wk old mouse (arrow). GFRA1+ tdTOMATO A_s cell (asterick). (**B'**) tdTOMATO is co-expressed with EOMES (top row, arrow). tdTOMATO is expressed in GFRA1+ A_s cells but not in chains of GFRA1+ A_al cells (bottom row). (**C**) tdTOMATO cells (Red), positive for GFRA1 (Green), are present within the seminiferous tubules. Top - An optical section of *Figure 3 continued on next page*

*Figure 3 continued*

confocal image showing the localization of EOMES+ cell along the basement membrane. Bottom – Magnified view of GFRA1+ EOMES [tdTOMATO+] cells shown above. Arrowheads mark myoid cell (M) nucleus, which are negative for EOMES [tdTOMATO]. * Background fluorescence in the interstitial space associated with GFRA1 staining. (D) Experimental flow chart showing the induction times and clonal analysis of *Eomes* [iCreERt2] targeted cells labeled in mice crossed to Rosa26 Confetti reporter and fed a tamoxifen-enriched diet. Fluorescence images of whole-mount seminiferous tubules showing the different size of labeled clones. Representative clones at 2 (n = 5, animals), 3 (n = 4), and 4wks (n = 3), and 4 (n = 3) and 7 (n = 3) months after tamoxifen induction. (E) Images of labeled sperm from the epididymis of mice 7 months after tamoxifen induced recombination. (F) Fluorescence positive offspring born to WT female mated with ROSA26confetti: *Eomes-iCreERt2* 4 months after tamoxifen-induced recombination at Confetti locus by *Eomes-iCreERt2*.

DOI: https://doi.org/10.7554/eLife.43352.008

The following figure supplement is available for figure 3:

**Figure supplement 1.** No TAM controls for lineage tracing of *Eomes* [iCreERt2] targeted cells in mice crossed to Rosa26 Confetti reporter on a normal (no tamoxifen) diet.

DOI: https://doi.org/10.7554/eLife.43352.009

(*Figure 3C* - top) and along the basement membrane juxtaposed to peritubular myoid cells (*Figure 3C* – bottom, arrowheads).

Next, we traced the progeny of *Eomes* [iCreERt2] cells during normal steady-state spermatogenesis after tamoxifen-induced recombination using the *Gt(ROSA) 26* [Sortm1(CAG-Brainbow2.1)Cle]/J (also referred to as *R26R-confetti*) reporter mice (*Madisen et al., 2010*). Mice were fed a tamoxifen-enriched diet for 2 weeks and clone sizes were determined by the number of cells expressing a single fluorescent protein after recombination at the *R26R-confetti* locus and detected by fluorescence imaging of whole mount seminiferous tubules (*Figure 3D*). One color clones indicate derivation from a single-labeled cell, or the overlap of clones expressing the same fluorescent marker. We observed confetti-marked single cells and chains of 4, 8 and 16 by two weeks (*Figure 3D*, top row). The size of labeled clones increased with time (*Figure 3D*, second and third row) and large labeled clones of differentiated germ cells were observed at 4 and 7 months after *iCre* activation (*Figure 3D*). Importantly, we observed single-labeled cells at all time points analyzed (*Figure 3D*, arrows), suggesting the single-labeled cells were either maintained for the entire duration or had undergone self-renewal and migrated away from their sister cells during the time the clones were traced. The labeled clones gave rise to labeled sperm detected in the epididymis of these mice (*Figure 3E*), and fluorescence-positive pups were born to *Eomes-[iCreERt2]: R26R-confetti* mice treated with tamoxifen when mated with WT C57BL6/J females (*Figure 3F*). Labeled clones, or offspring, were not observed in the absence of TAM at any time points (*Figure 3—figure supplement 1*). Together, these results demonstrate that EOMES-expressing cells contribute to long-term maintenance of $A_{single}$ spermatogonia and to full-lineage differentiation to spermatozoa in mice.

## EOMES+ cells are resistant to busulfan treatment

To determine if EOMES+ cells are capable of regeneration after chemical insult, we specifically killed proliferating spermatogonia with busulfan and observed how EOMES+ and EOMES− cells survived the insult. We chose a low dose (10 mg/kg body weight) that preferentially kills proliferating $A_{pr}$ and $A_{al}$ spermatogonia (*Keulen and Rooij, 1973*). Seminiferous tubules from treated mice were immunostained for EOMES and PLZF every 3 days between 3 and 15 days after busulfan injection (*Figure 4A*). During the entire 15 day time period the EOMES+ population remained unchanged (*Figure 4B*). In contrast, by 9 days after treatment, the PLZF+ population had halved, presumably due to loss of $A_{pr}$ and $A_{al}$ cells. The population began to recover after 12 days (*Figure 4B*). We also stained for EOMES and GFRA1. At day 0,~20% of the total GFRA1-expressing cell population was EOMES+ (*Figure 4C*). Comparison within the GFRA1+ population showed a selective increase of the EOMES+ fraction after busulfan treatment (*Figure 4C*), suggesting that the EOMES− fraction was specifically lost.

To assess if EOMES+ cells contribute to regeneration after chemical injury, 3wk old *Rosa26confetti: Eomes-[iCreERt2]* mice were placed on a TM diet for 12 days to activate CreERt2, treated with a low dose of busulfan (10 mg/kg body weight) on day 12, and then analyzed for labeled clones on day 45 (*Figure 4D*). Whole mount seminiferous tubules showed labeled clones at varying stages of differentiation, supporting the conclusion that EOMES+ GFRA1+ PLZF+ spermatogonia are resistant

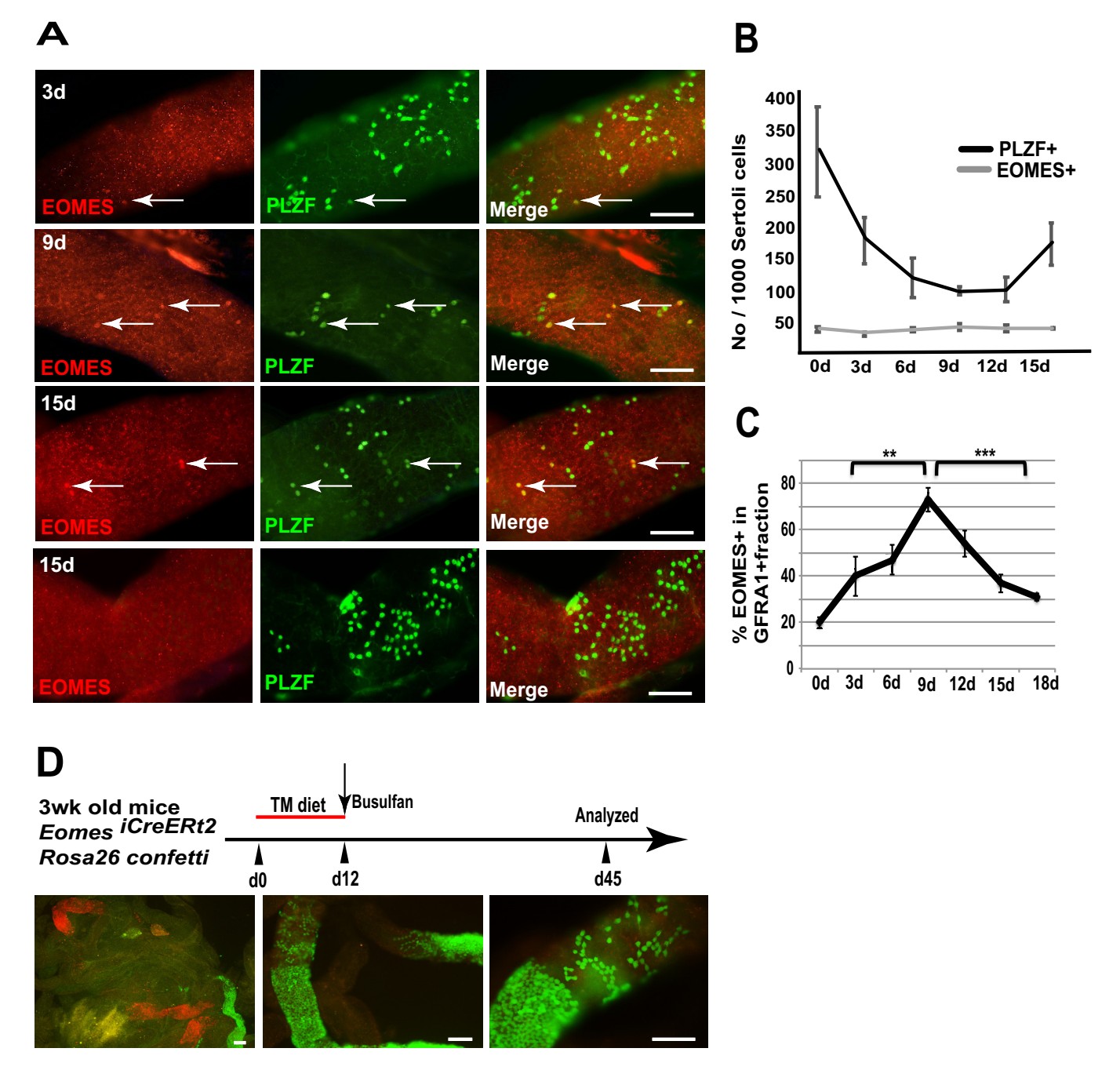

**Figure 4.** EOMES+ cells are resistant to busulfan and contribute in regeneration. (A) Whole-mount tubule immunostaining for EOMES and PLZF after low-dosage busulfan treatment (10 mg/mL). (B) Cell counts from (A) show that EOMES+ cell numbers remain constant while PLZF+ cell numbers drop after busulfan treatment. (n = 3 per time point, 8 weeks old) (C) The EOMES+ fraction of GFRA1+ cells increased following busulfan treatment. **, p<0.005; ***, p<0.0001. (D) Schedule to assay the regeneration properties of EOMES+ cells. Three-week old *R26R-confetti: Eomes* $^{iCreERt2}$ mice were put on a tamoxifen-enriched diet for 1 week, treated with busulfan, and analyzed on day 45. Images of whole mount seminiferous tubules (n = 5 mice) show labeled clones at different developmental stages during regeneration.

DOI: https://doi.org/10.7554/eLife.43352.010

The following source data is available for figure 4:

**Source data 1.** Source file for *Figure 4B and 4C*.

DOI: https://doi.org/10.7554/eLife.43352.011

to busulfan insult compared to their EOMES− counterparts, and that the EOMES+ population reconstitutes the lineage after chemical injury.

## Loss of PLZF increases the proliferative rate of EOMES+ cells

We speculated that the progressive age-dependent loss of germ cells in the *Plzf* $^{lu/lu}$ mutants could be due to an alteration in SSC cycling rates resulting in SSC exhaustion. To test this hypothesis, we injected EdU into WT and *luxoid* mice, and quantified the incorporation rate of EdU in EOMES+ cells. Although fewer EOMES+ cells were present in *lu/lu* testes compared to WT, a greater fraction of the EOMES+ cells had incorporated EdU compared to WT (36% vs. 10%, *Figure 5A and B*). These results indicate that EOMES+ cells have a higher than normal proliferative index in the absence of PLZF, suggesting that PLZF directly, or indirectly, regulates the proliferation of EOMES+ cells.

To determine if the fraction of cycling EOMES+ cells changes during regeneration, mice were injected with EdU at 9 or 12 days post busulfan injection and assayed by immunofluorescence in whole mount seminiferous tubules. At both time points the EdU labeling index was increased in EOMES+ cells (*Figure 5C*) suggesting that busulfan is either directly inducing proliferation, or the cells are proliferating in response to a decrease in SSC density or a shortage of differentiating cells (*Rooij et al., 1985*).

## Single cell RNA sequencing of EOMES $^{tdTOMATO+}$ cells

The expression or function of a growing collection of genes have been linked to either SSC self-renewal or progenitor cell expansion (*Table 1*). To address if these SSC identity genes are expressed in EOMES+ cells, we performed single cell RNA sequencing (scRNAseq) on 897 fluorescently activated cell sorted (FACS) EOMES $^{tdTOMATO+}$ cells from 4wk old *Eomes-iCreERt2-tdTomato; Plzf* $^{+/+}$ and 352 EOMES $^{tdTOMATO+}$ cells from *Eomes-iCreERt2-tdTomato; Plzf* $^{lu/lu}$ mice (*Figure 6A*). The mean number of expressed genes per cell in *Plzf* $^{+/+}$ and *Plzf* $^{lu/lu}$ was 3104 and 3230, while the mean number of transcripts with unique molecular identifiers was 7254 and 8134, respectively (*Figure 6—figure supplement 1*).

An expression heat-map of the identity genes indicates that the majority of the cells express *Gfra1*, *Bmi1*, *Id4* and *Plzf*, while fewer cells express *T*, *Eomes*, *Pdx1*, *Nanos2*, *Neuorg3* and *Pax7* (*Figure 6B*). The most striking finding is the considerable degree of heterogeneity in expression across the entire scRNAseq data set. None of the cells expressed all of the genes, while 90% (807/897) of cells expressed at least one of the genes and 73% (655/897) expressed two or more of the genes. *Gfra1* was detected in 566/897 cells (63%), while PAX7 was detected in only 10/897 cells (1.11%). *Neurog3* (*Yoshida et al., 2004*), which is expressed in early progenitor cells, was detected at relatively low levels in 27/897 cells (3.01%), suggesting that *Neurog3* is expressed in some SSCs, or that the sorted cells include early progenitor cells due to a longer half-life of tdTOMATO compared to EOMES itself. *Eomes* transcripts were detected in 85/897 (9.48%) EOMES $^{tdTOMATO+}$ cells. The low number of cells could be due to the inherent low expression of *Eomes* relative to other expressed genes, early progenitor cells that are tdTOMATO positive but are no longer expressing *Eomes* transcript, or the cell cycle stage in which the cells were collected. Together, the heterogeneity in transcript detection in the scRNAseq data mirrors previously reported heterogeneity in protein expression as detected by immunofluorescence (*Suzuki et al., 2009*).

To determine the relationship between the cells expressing the identity genes, cluster analysis was performed on the 100 most differentially expressed genes, plus 10/11 identity genes, from *Plzf* $^{+/+}$ sorted cells. *Pax7* was excluded from the analysis because its normalized mean expression was too low to be meaningful. The bulk of the cells clustered together (cluster 2) as a group of 740 cells (*Figure 6C*). The remaining cells were defined by a separate group of 5 different smaller clusters. Cells assigned to cluster 0 could not be confidently assigned to any cluster. The expression of *Dazl*, a marker of germ cells (*Cooke et al., 1996*), at moderate to high levels in all clusters, confirms that each cluster represents germ cells (*Figure 6C*). The expression of *Ldhc*, which is highly restricted to germ cells and whose protein is first detected in early pachytene spermatocytes (*Goldberg et al., 2010*), is expressed at highest levels in clusters 3–5, which are represented by a few cells, suggesting that *Ldhc* mRNA is expressed at low levels in SSCs or early progenitor cells. Although *Gfra1* was detected in all clusters, its highest mean expression was in cluster 1, as was *Id4* and *Bmi1* (*Figure 6D*). Although cluster 1 contained only 13 cells, *Eomes*, *T*, *Pdx1*, and *Nanos2* were

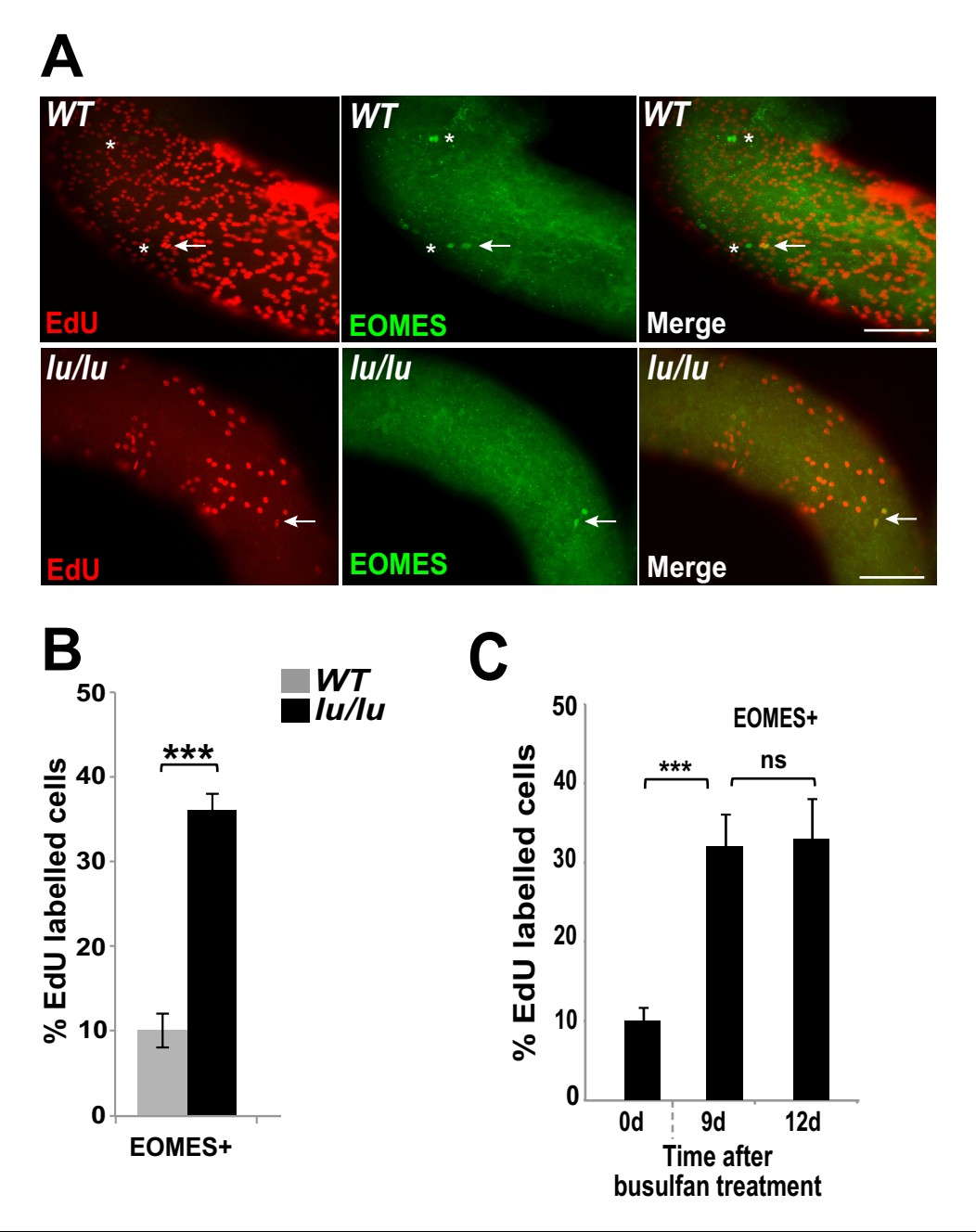

**Figure 5.** Loss of PLZF increases the proliferative index of EOMES+ cells. (**A**) Whole-mount seminiferous tubules from 8 week old WT and *lu/lu* mice injected with EdU and immunostained 24 hr after injection. EdU+ EOMES+ cells (arrows); EdU- EOMES+ cells (asterisks). (**B**) Quantification of the EdU labeling index in WT (n = 4 mice) and *lu/lu* (n = 5 mice) EOMES+ cells. EOMES+ cells have a higher labeling index in *lu/lu* than in WT (p=0.001). (**C**) Increase in proliferation rate of EOMES+ cells during the recovery phase after busulfan treatment. EdU incorporation was assayed 24 hr post EdU injection (n = 3 mice per timepoint; ***p=0.001).

DOI: https://doi.org/10.7554/eLife.43352.012

The following source data is available for figure 5:

**Source data 1.** Source file for *Figure 5B and 5C*.

DOI: https://doi.org/10.7554/eLife.43352.013

**Table 1.** Key identity genes expressed in SSCs or progenitor cells.

| Gene | Expression | Method | Functional Assay | References |
|------|-----------|--------|------------------|-----------|
| *Gfra1* | $A_s$, $A_{pr}$, $A_{al(4-8)}$ | Gfra1-$^{CreERT2}$ | Lineage tracing Transplant | (*Hara et al., 2014*) |
| *Bmi1* | $A_s$ | Bmi1$^{CreERT}$ | Lineage tracing | (*Komai et al., 2014*) |
| *Id4* | $A_s$ | Id4-$^{eGFP}$ | Lineage tracing Transplant | (*Oatley et al., 2011*; *Sun et al., 2015*) |
| *Plzf* | $A_s$, $A_{pr}$, $A_{al(4-16)}$ | Antibody staining | Lineage tracing Transplant | (*La et al., 2018*; *Buaas et al., 2004*) |
| *Lin28a* | $A_s$, $A_{pr}$, and $A_{al(4-16)}$ | Antibody staining | Transplant | (*Chakraborty et al., 2014*) |
| *T* | $A_s$ | T$^{(nEGFP-CreERT2)}$ | Lineage tracing | (*Tokue et al., 2017*) |
| *Eomes* | $A_s$, $A_{pr}$ | Eomes-$^{iCre\ ERt2\ tdTomato}$ | Lineage tracing | This manuscript |
| *Nanos2* | $A_s$, $A_{pr}$ | Antibody staining | Lineage tracing Knockout | (*Sada et al., 2009*) |
| *Neurog3* | $A_{al(4-16)}$ | Ngn3$^{CreERT}$ | Lineage tracing Transplant | (*Nakagawa et al., 2010*) |
| *Pax7* | $A_s$ | Pax7-$^{CreERT2}$ | Lineage tracing | (*Aloisio et al., 2014*) |
| *Pdx1* | $A_s$, $A_{pr}$ | Antibody staining | Lineage tracing | (*La et al., 2018*) |

DOI: https://doi.org/10.7554/eLife.43352.016

expressed in a higher proportion of cells (e.g. 23–38% in cluster 1 compared to 9–15% in cluster 2) than in any other cluster. Nonetheless, the presence of multiple clusters expressing different identity genes suggests that EOMES $^{tdTOMATO+}$ sorted cells represent a collection of cells with different transcriptional profiles and supports the conclusion that SSCs are transcriptionally heterogeneous.

To determine the impact of loss of *Plzf* on the expression of the identity genes, we computed the change in expression of each gene across all cells in *Plzf* $^{+/+}$ and *Plzf* $^{lu/lu}$ mice. In *Plzf* $^{lu/lu}$ mice, the $\log_2$-fold change in expression of the identity genes *Bmi1*, *Id4*, *Pax7*, *Nanos2* and *Neurog3* were unchanged compared to *Plzf* $^{+/+}$ mice, while the expression of *Gfra1*, *Eomes*, *T*, and *Pdx1* were significantly decreased (*Figure 6B* and *Table 2*). In *Plzf* $^{lu/lu}$ mutants, the fraction of cells expressing *Gfra1* was reduced from 63.10% in *Plzf* $^{+/+}$ cells to 38.07% in *Plzf* $^{lu/lu}$ mutants. *Eomes* and *T* were only expressed in (1.14%) of *Plzf* $^{lu/lu}$ mutant cells, compared to 9.48% and 14.94%, in *Plzf* $^{+/+}$ cells, respectively, suggesting that mutation of *Plzf* differentially affects a subpopulation of SSCs expressing *Gfra1*, *T*, *Pdx1*, and *Eomes*.

Given that ID4-EGFP has previously been reported to be restricted to a subpopulation of $A_s$ cells (*Helsel et al., 2017*), that it has been proposed to mark the ultimate pool of SSCs (*Helsel et al., 2017*), and that we did not detect significant changes in *Id4* transcript in EOMES $^{tdTOMATO+}$ sorted cells in *Plzf* $^{lu/lu}$ mutants, we investigated the expression of ID4 by immunofluorescence. Using a rabbit polyclonal antibody, which detects a protein of the correct molecular weight by western blotting in extracts prepared from WT testes but not in extracts from *Id4*$^{-/-}$ testes (*Figure 7A*), we detected ID4 in PLZF+ $A_s$, $A_{pr}$, and $A_{al}$ cells (*Figure 7C*). Because this staining pattern differed from that previously reported for an *Id4-EGFP* transgene, we performed double-staining for ID4 and LIN28A. We again found that we could detect ID4 in chains of $A_{al}$ cells that were also positive for LIN28A (*Figure 7D*). ID4 immunostaining was not detected in seminiferous tubules of *Id4* $^{-/-}$ mutants (*Figure 7E*) (*Yun et al., 2004*). We assayed for co-expression of tdTOMATO and ID4 and found that 40% of the tdTOMATO+ $A_s$ cells were also positive for ID4 (*Figure 7F*, upper row). We again detected ID4 in chains of $A_{al}$ cells (*Figure 7F*, bottom row), none of which were positive for tdTOMATO. In aggregate, we determined that only 20% of ID4+ cells were positive for tdTOMATO. To determine if *Id4* was required for expression of EOMES, we performed immunofluorescence on testis from *Id4*$^{-/-}$ mutants. EOMES was detected in GFRA1+ cells (*Figure 7G*), consistent with the detection of *Eomes* RNA by rtPCR in *Id4*$^{-/-}$ mutants (*Figure 7B*). We conclude that ID4 is expressed in a subpopulation of EOMES+ cells, but the majority (80%) of the ID4+ population is EOMES-.

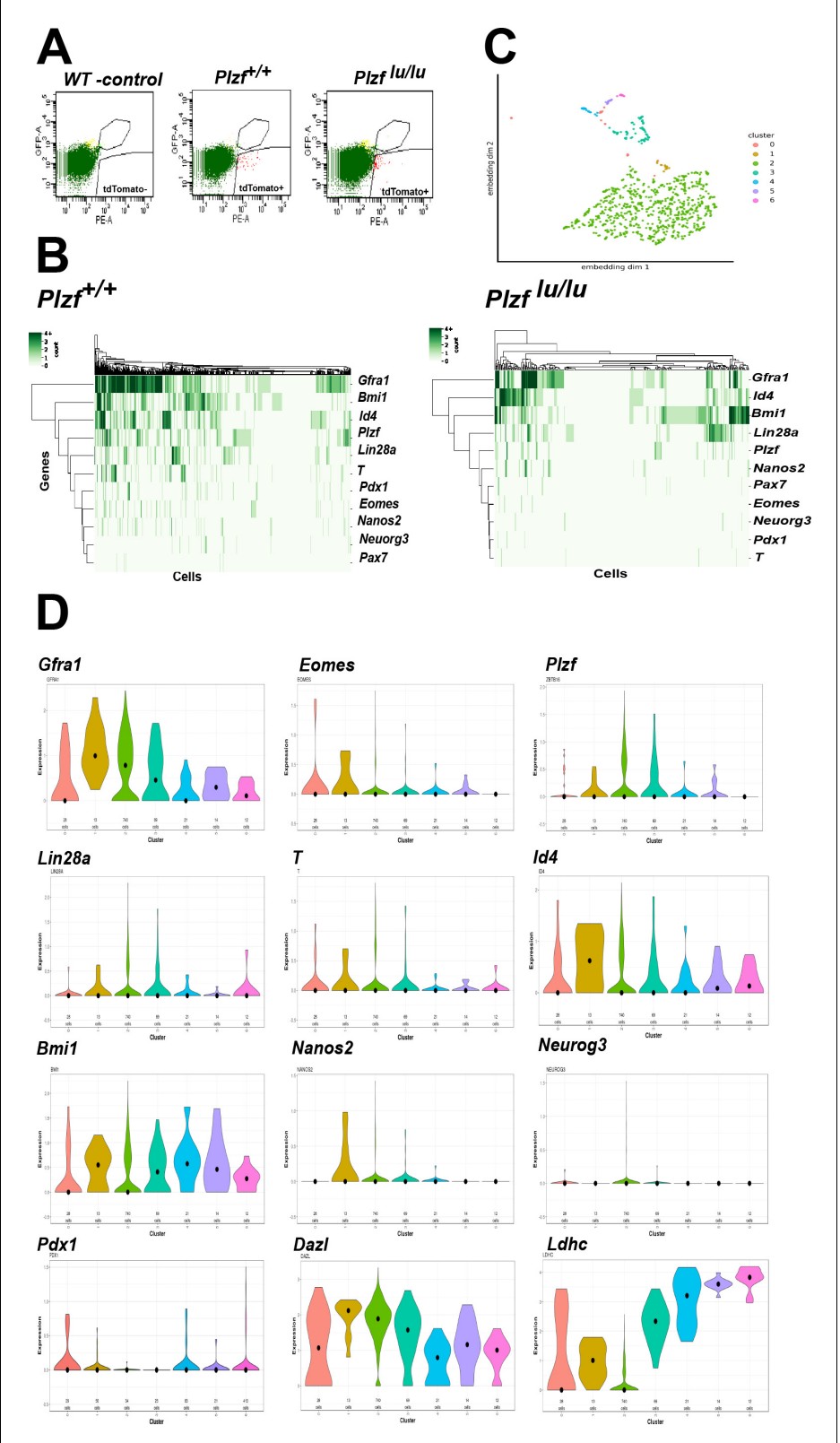

**Figure 6.** Single cell RNA sequencing of EOMES [tdTOMATO] cells. (A) FACscan profile of EOMES [tdTOMATO+] cells sorted from *Plzf* [+/+] and Plzf [lu/lu] mutant testis. The boxed region in the upper right indicated cells with background autofluorescence that were excluded from the analysis. (B) Heat map and dendogram of SSC identity and progenitor genes expressed in EOMES [tdTOMATO] cells from *Plzf* [+/+] and Plzf [lu/lu] mutant testis. (C) Clusters

*Figure 6 continued on next page*

*Figure 6 continued*

generated from WT cells. Cellular expression profiles of 110 genes were embedded into a 2dimensional latent space using UMAP (see supplemental information for details). Cells in cluster 0 could not be confidently placed in any of the other six clusters. (D) Violin plots showing the mean expression of each SSC identity gene per cluster in cells sorted from *Plzf* $^{+/+}$ testes.

DOI: https://doi.org/10.7554/eLife.43352.014

The following figure supplement is available for figure 6:

**Figure supplement 1.** Distribution of unique molecular identifiers (UMIs) and genes per cell.

DOI: https://doi.org/10.7554/eLife.43352.015

## Discussion

Exploiting a transgenic overexpression model for GDNF in *Plzf* $^{+/+}$ and *Plzf* $^{lu/lu}$ mice, we discovered a rare population of $A_s$ EOMES-expressing spermatogonia. Lineage tracing studies during steady state spermatogenesis and during regeneration demonstrated that EOMES+ cells are SSCs. The significant loss of *Eomes*-expressing cells between *Plzf* $^{+/+}$ and *Plzf* $^{lu/lu}$ suggest that EOMES marks a subpopulation of GFRA1+ cells that are uniquely depleted in *Plzf* $^{lu/lu}$ mice. EOMES+ cells cycle more frequently in *Plzf* $^{lu/lu}$ mice, suggesting that PLZF regulates their proliferative activity.

In addition to expression in $A_s$ cells, EOMES was occasionally detected in one of two adjacent GFRA1+ cells. The clones, which we counted as $A_{pr}$, could have also been two lineage-independent, but adjacent, $A_s$ cells. Alternatively, the two cells could be daughter cells, suggesting that EOMES expression can be asymmetrically inherited. Interestingly, in rare cases EOMES was expressed heterogeneously in a chain of GFRA1+ $A_{al}$ cells, again suggesting that the fate of cells with clones of $A_{al}$ is not uniform and supporting the possibility that cells within chains are subject to reversion to SSCs following fragmentation (*Nakagawa et al., 2010*).

Single cell sequencing of EOMES$^{tdTOMATO+}$ cells revealed extensive heterogeneity of expression of a collection of SSC identity genes (*Figure 6*). Immunostaining of several of the markers confirmed that SSCs are heterogeneous yet hierarchical, with markers such as EOMES preferentially expressed in more primitive $A_s$ cells and LIN28A expressed in differentiating $A_s$, $A_{pr}$ and $A_{al}$ cells (*Figure 2H*). Similar conclusions from both protein and RNA expression studies have been reported by others (*Suzuki et al., 2009*; *La et al., 2018*).

Overexpression of GDNF led to clusters of tightly packed nests of undifferentiated spermatogonia. Cells in the center of the nests tended to be EOMES+ LIN28 A- while cells at the periphery were EOMES- LIN28A+. We speculate that the significance of the geometry of the nests could be a consequence of increased paracrine signaling, perhaps driven by intercellular coupling of GDNF with its co-receptor GFRA1/RET, or the physiological state in the center of the nests, that favors SSC self-renewal.

**Table 2.** Differential expression of SSC identity transcripts in Plzf +/+ and Plzf lu/lu mutant cells.

| Gene | Log2 FC | P value |
| --- | --- | --- |
| Gfra1 | -0.779 | 1.31E-12 |
| Eomes | -0.766 | 1.41E-09 |
| Plzf | -1.27 | 4.09E-21 |
| Id4 | -0.046 | 0.73 |
| Bmi1 | -0.021 | 0.8 |
| Pax7 | 0.09 | 0.278 |
| T | -1.45 | 2.11E-19 |
| Lin28a | 0.046 | 0.71 |
| Nanos2 | 0.039 | 0.67 |
| Neurog3 | -0.177 | 0.08 |
| Pdx1 | -0.738 | 6.11E-09 |

DOI: https://doi.org/10.7554/eLife.43352.017

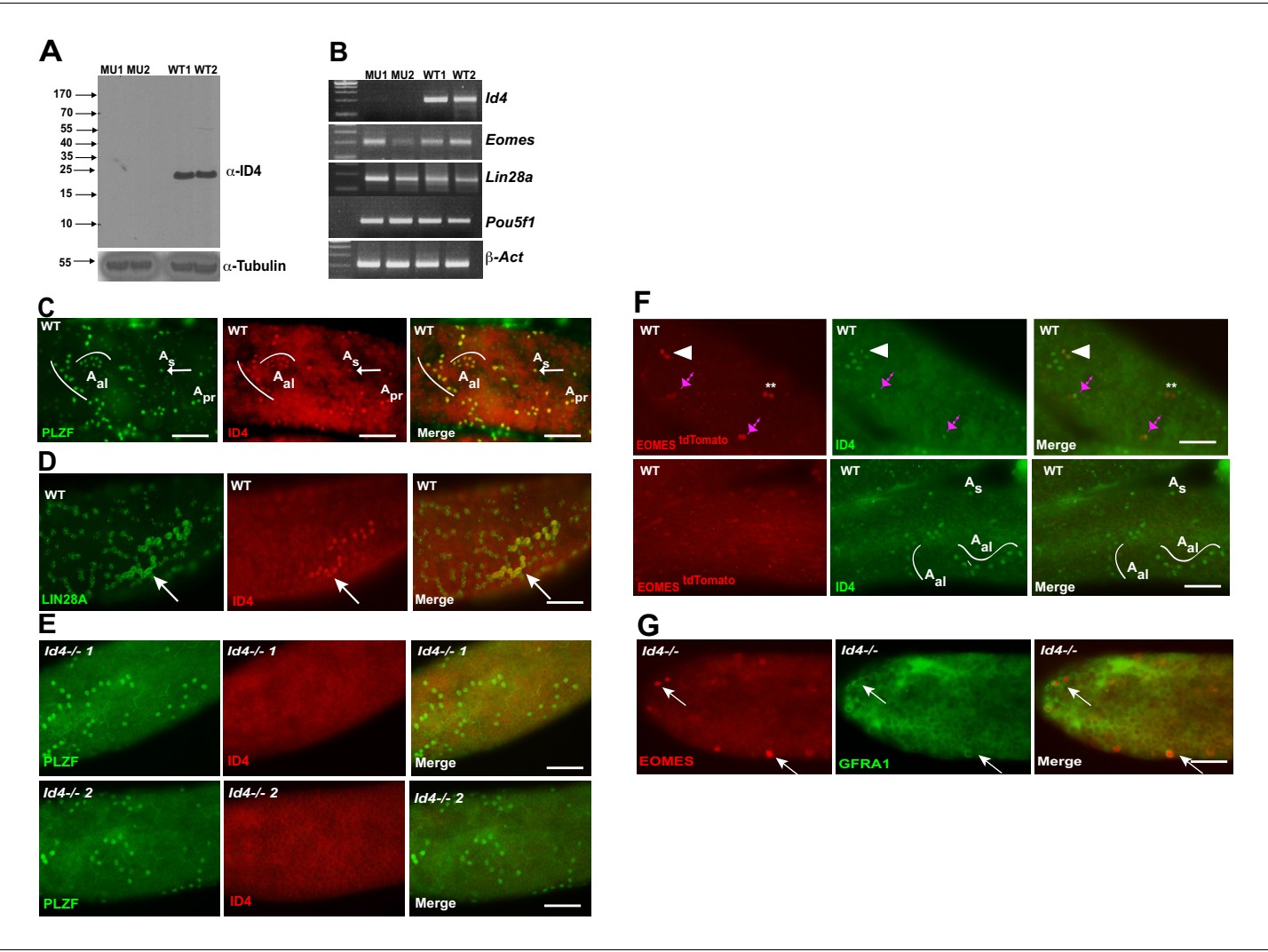

**Figure 7.** ID4 is expressed in A$_s$, A$_{pr}$ and A$_{al}$ spermatogonia. (**A**) Western blot of total testis protein lysate from 2 WT and 2 *Id4$^{-/-}$* mutant mice probed with ID4 antibody. For loading control, western blot was probed with α-tubulin antibody. (**B**) RT-PCR using total RNA from two *Id4$^{-/-}$* mutant and 2 WT mice for *Eomes, Lin28a, Pou5f1 and β-actin*. (**C**) Detection of PLZF and ID4 by immunofluorescence in whole mount seminiferous tubules. Both ID4 in PLZF+ were detected in A$_{s\,(arrow)}$, A$_{pr}$, and A$_{al}$ spermatogonia in WT adult mouse testis. (**D**) ID4 protein was co-localized in chains of LIN28A+ A$_{al}$ spermatogonia (arrow). (**E**) ID4+ cells are not detected in whole mount tubules from prepared from *Id4$^{-/-}$* mutant mice. (**F**) Top row, examples of tdTOMATO/ID4 double-positive cells (arrowhead), tdTOMATO+ ID4- (**), and two adjacent cells where one is tdTOMATO+ and the other is tdTOMATO/ID4 double-positive (magenta arrow). Bottom row, examples of tdTOMATO- ID4+ A$_s$ and A$_{al}$ chains of 4 and 8 cells. (**G**) EOMES+ GFRA1 + cells are present in *Id4$^{-/-}$* mutants (arrow).

DOI: https://doi.org/10.7554/eLife.43352.018

## Relation of EOMES cells to other SSC markers

Single-cell RNA sequencing of EOMES $^{tdTOMATO+}$ cells from *Plzf $^{lu/lu}$* mutants suggests that EOMES-expressing cells constitute a unique population of A$_s$ cells. In recent studies, other GFRA1+ A$_s$ sub-populations have also been identified. A recent report supports our data that a subset of A$_s$ spermatogonia are marked by EOMES, T, PDX1 and LHX1, and transplantation studies have shown that PDX1-expressing cells are SSCs (*La et al., 2018*). A *Bmi1 $^{CreERT}$* allele has also been shown to mark SSCs (*Komai et al., 2014*). We detected *Bmi1* transcript in EOMES $^{tdTOMATO+}$ cells, and long-term labeling of SSCs has been observed in *Bmi1* lineage tracing experiments, however, the mean expression of *Bmi1* was similar between *Plzf $^{+/+}$* and *Plzf $^{lu/lu}$* cells, suggesting that *Bmi1* expression is not restricted to EOMES+ cells. *Pax7* lineage-marked cells are capable of repopulating germ cell-free testes in transplants and are resistant to busulfan insult. (*Aloisio et al., 2014*). However, *Pax7*

transcript was detected in only 10/897 EOMES $^{tdTOMATO+}$ cells, and its mean expression too low to be included in our cluster analysis.

An *Id4-gfp* allele also marks a subset of A$_s$ spermatogonia (*Chan et al., 2014*; *Oatley et al., 2011*; *Sablitzky et al., 1998*; *Sun et al., 2015*) and recent studies have shown that transplantable SSC activity resides in the ID4-GFP-bright population (*Helsel et al., 2017*). Because ~40% of EOMES + cells are ID4+, and *Id4* transcripts were co-expressed in 49 of the 85 *Eomes+* cells, it is likely that there is substantial overlap in the population of cells marked by EOMES, ID4-GFP-bright, and T (47% of cells that expressed *Eomes* also expressed *T*) (*Tokue et al., 2017*). In fact, previous studies have shown that there is a 4.15 $\log_2$ fold (p=0.00005) enrichment of *Eomes* transcript in the ID4-GFP-bright population (source file Database S1 in *Helsel et al., 2017*), suggesting a strong overlap in EOMES-expressing cells and the ID4 bright population. However, our data suggest that the *Id4-gfp* transgene may not reflect the full expression pattern of the ID4 protein itself. Although we detected ID4 in ~40% of the EOMES+ cells, the majority of the ID4+ cells (80%) were EOMES- and included A$_{pr}$ and A$_{al}$ cells. Differences in half-lives of the ID4 and EGFP proteins, or differences in translational control elements in their respective mRNAs, may be responsible for the discordance in their expression. Others have also recently reported that expression of *Id4$^{IRES-Gfp}$* is not restricted to A$_s$ cells and that approximately 45% of ID4-IRES-GFP+ cells are EOMES- and PDX1- spermatogonia, confirming that ID4 expression is not restricted to A$_s$ cells (*La et al., 2018*).

At face value our results are inconsistent with the fragmentation model for SSC self-renewal (*Nakagawa et al., 2010*; *Hara et al., 2014*). However, it is possible that the *Ngn3* and *GFRa1* lineage tracing studies preferentially labeled the equivalent of our EOMES- GFRA1+ SSCs, and that EOMES- A$_s$, A$_{pr}$ and A$_{al}$ cells can de-differentiate to EOMES+ A$_s$ SSCs following fragmentation and transplantation. Interestingly, PDX1-expressing cells are lost upon busulfan ablation but then regained during regeneration (*La et al., 2018*).

## Function of PLZF

To date, it is unclear how PLZF mechanistically regulates SSC self-renewal, as it is expressed in the entire undifferentiated spermatogonia population. Our data demonstrate that PLZF regulates the cycling status of EOMES+ cells, and suggests that the failure of *luxoid* mutant germ cells to successfully colonize after transplantation is a consequence of their severely depleted population of EOMES + A$_s$ cells and the altered proliferative index of the entire population of GFRA1+ cells. This paradigm could extend to the regulation of germline stem cells in humans, as PLZF is expressed in human SSCs (*Hermann et al., 2010*). Several studies have provided in vitro evidence for a role of PLZF in cell cycle regulation. PLZF associates with CDC2 kinase activity in vitro (*Ball et al., 1999*) and when overexpressed in myeloid progenitors, PLZF decreases proliferation by reducing entry into S-phase (*Doulatov et al., 2009*). Furthermore, PLZF suppresses growth of 32Dcl3 cells in vitro by repressing transcription of the cyclin A2 gene and other cyclin-dependent complexes involved in the G$_1$/S transition (*Shaknovich et al., 1998*; *Yeyati et al., 1999*). Our studies show that PLZF is critical for maintaining SSCs in a low proliferative state and suggest that it regulates the cycling status of SSCs. Analogous to the hematopoietic system, which contains both slow-cycling long-term (LT)-HSCs, and rapid-cycling short-term HSCs, (*Cheshier et al., 1999*; *Morrison and Weissman, 1994*; *Spangrude et al., 1988*), and where over-cycling of LT-HSCs can lead to stem cell exhaustion and consequent loss of differentiated cell types (*Arai and Suda, 2007*), we propose that loss of PLZF results in proliferative exhaustion of SSCs.

## Materials and methods

**Key resources table**

| Reagent type (species) or resource | Designation | Source or reference | Identifiers | Additional information |
|---|---|---|---|---|
| Strain, strain background (*Mus musculus*) | B6/N | The Jackson Laboratory | JR # 05304 | |

*Continued on next page*

*Continued*

| Reagent type (species) or resource | Designation | Source or reference | Identifiers | Additional information |
|---|---|---|---|---|
| Strain, strain background (*Mus musculus*) | *B6.129P2-Gt(ROSA) 26* $^{Sortm1}$ $_{(CAG-Brainbow2.1)Cle}$/J | The Jackson Laboratory | JR# 017492 | |
| Strain, strain background (*Mus musculus*) | *Zbtb16$^{lu}$/J (Plzf $^{lu}$)* | The Jackson Laboratory | JR#000100 | |
| Strain, strain background (*Mus musculus*) | B6(Cg)-$^{Tyr\ c-2J}$/J | The Jackson Laboratory | JR# 000058 | |
| Strain, strain background (*Mus musculus*) | B6.Cg-*Gt(ROSA) 26Sor* $^{tm6(CAG-ZsGreen)Hze}$/J | The Jackson Laboratory | JR# 007906 | |
| Strain, strain background (*Mus musculus*) | Tg(Ctsl-Gdnf) B6.C3H.F1 | Sharma & Braun 2018 | PMCID: 29440301 | |
| Strain, strain background (*Mus musculus*) | *Id4* $^{-/-}$ | Received from Mathew Havrda and Mark Israel; Dartmouth Medical School | PMCID: 15469968 | |
| Cell line (*M. musculus*) | B6N-JM8 ES | The Jackson Laboratory | RRID:CVCL_J957 | |
| Antibody | Mouse mono-clonal anti-PLZF | Santa Cruz Biotech Inc, Santa Cruz, CA | Cat# sc-28319 | Dilution – 1:250 |
| Antibody | Goat poly-clonal anti-GFRA1 | R and D systems, Minneapolis, MN | Cat# AF560 | Dilution – 1:500 |
| Antibody | Rat poly-clonal anti-LIN28A | Gift from Dr. Eric G. Moss, Rowan University, School of Osteopathic Medicine, One Medical Center Drive, Stratford, NJ 08084 | | Dilution – 1:500 |
| Antibody | Rabbit poly-clonal anti-EOMES | Abcam, Cambridge, MA | Cat# ab23345 | Dilution – 1:500 |
| Antibody | Rabbit polyclonal anti-SOHLH1 | Abcam, Cambridge MA | Cat# ab49272 | Dilution – 1:400 |
| Antibody | Rabbit mono-clonal anti-ID4 | BioCheck Inc, Foster City, CA | Cat # BCH-9/ # 82–12 | Dilution – 1:500 |
| Chemical compound, drug | Tamoxifen diet | https://www.envigo.com/ products-services/teklad/ laboratory-animal-diets/ | TD. 130859 | |
| Chemical compound, drug | EdU | Invitrogen, https://www.thermofisher. com/us/en/home/brands/ invitrogen.html | Cat# A10044 | |
| Chemical compound, drug | Busulfan | Sigma-Aldrich, St. Louis, MO | Cat# B2635 | |
| Commercial assay or kit | Click-IT EdU assay kit | Invitrogen https://www.thermofisher. com/us/en/home/brands/ invitrogen.html | Cat # A20185 | |
| Commercial assay or kit | RNA isolation PureLink RNA Mini kit | Ambion https://www.thermofisher. com/us/en/home/brands/ invitrogen/ambion.html | Cat# 12183018A | |

*Continued on next page*

*Continued*

| Reagent type (species) or resource | Designation | Source or reference | Identifiers | Additional information |
|---|---|---|---|---|
| Commercial assay or kit | Arcturus PicoPure RNA kit | Applied Biosystems https://www.thermofisher.com/us/en/home/brands/applied-biosystems.html | Cat # 12204–01 | |
| Commercial assay or kit | TruSeq RNA Sample Prep Kit v2 | Illumina https://www.illumina.com/ | | |
| Commercial assay or kit | Nextera Xt DNA sample kit | Illumina https://www.illumina.com/ | | |
| Sequence-based reagent | Forward primers for RT-PCR of *Eomes* | | | 5'- ctggttctgttctttgcacaggcgcatg-3' |
| Sequence-based reagent | Reverse primers for RT-PCR of *Eomes* | | | 5'- atcagccacaccagacacagagatc-3' |
| Sequence-based reagent | Forward primers for RT-PCR B-actin | | | 5'- ccagttcgccatggatgacgatat-3' |
| Sequence-based reagent | Reverse primers for RT-PCR B-actin | | | 5'- gtcaggatacctctctgctctga-3' |
| Software, algorithm | TopHat v 2.0.7 | *Kim et al., 2013* | PMCID:PMC4053844 | |
| Software, algorithm | EdgeR v2.6.10 | *Robinson et al., 2010* | PMCID:PMC2796818 | |
| Software, algorithm | BamTools v 1.0.2 | *Barnett et al., 2011* | PMCID:PMC3106182 | |
| Software, algorithm | Cell View | *Bolisetty et al., 2017* | http://doi.org/10.1101/123810 | |

## Animals

All animals were housed in a barrier facility at The Jackson Laboratory. All experimental procedures were approved by the Jackson Laboratory Institution Animal Care and Use Committee (ACUC) and were in accordance with accepted institutional and government policies.

## Generation of tamoxifen inducible *iCre ERt2-tdTomato* knock-in into the *Eomes* locus

The BAC clone RP23-235G22 was used to clone the *2A:CreERt2-FRT-PGK-Neor-FRT* cassette into exon 2 of mouse *Eomes* gene. Linearized vector was electroporated into B6N-JM8 ES cells from B6N mouse strain. Chimera were generated using B6(Cg)-$^{Tyr\ c-2J}$/J the albino mice. The chimeras were crossed to B6N parental strain to generated F1 progeny. The reporter mice used to assay for Cre recombination were *B6.129P2-Gt(ROSA) 26*$^{Sortm1(CAG-Brainbow2.1)Cle}$/J. For activation of *iCre* the mice were fed TD.130859 (TAM diet). The food was formulated for 400 mg tamoxifen citrate per kg diet that would provide ~40 mg tamoxifen per kg body weight per day.

## Testes weight and histology

Mice were genotyped with specific primer sets and assessed for body weight and testes weight at each time point analyzed. For histological analysis, testes were fixed overnight in Bouin's and paraffin-embedded sections were stained with PAS after dewaxing.

## Whole mount immunostaining

Seminiferous tubules were dispersed in PBS by removing the tunica from the testes. They were rinsed with PBS and fixed in 4% paraformaldehyde (Electron Microscopy Sciences, Fort Washington, PA) for 4–6 hr at 4℃. For PLZF, LIN28A, and SOHLH1 antibodies, the tubules were permeabilized with 0.25% NP-40 in PBS-T (0.05% Tween in PBS) for 25 min at RT before blocking in 5% normal

goat serum. The tubules were then incubated with primary antibodies and incubated overnight (GFRA1 and EOMES) at RT or 4°C for PLZF, LIN28, and SOHLH1. The next day the tubules were washed three times with PBS-Tween20 for 5 min, and incubated with secondary antibodies conjugated to Alexa Fluors (Molecular Probes) for 1 hr at room temperature. After washing in PBS-Tween20, the tubules were mounted in VectaShield with DAPI (Vector Laboratories, Burlingame, CA) and imaged using a Nikon Eclipse E600 epifluorescence microscope equipped with an EXi Aqua Camera from Q-Imaging (Surrey, BC, Canada).

## Proliferation assay

To assay the proliferative index of spermatogonial stem cells, mice were injected intra-peritoneally with 50 mg/kg body weight of EdU. EdU staining was performed using the Click-iT EdU imaging Kit (Invitrogen, Carlsbad, CA) according to manufacturer's protocol. Briefly, seminiferous tubules were dissected 24 hr after EdU injection and first probed with an EOMES antibody before staining for EdU. Tubules were incubated with Click-iT reaction cocktail containing buffer, $CuSO_4$, Alex Fluor 594 Azide, and reaction additive buffer for 30 min at room temperature. Tubules were then washed in PBS-Tween20 buffer for more than 2 hr and then mounted in Vectashield mounting media (Vector Laboratories Inc, Burlingame, CA) with DAPI and imaged on a Nikon Eclipse E600 epifluorescence microscope equipped with an EXi Aqua Camera from Q-Imaging (Surrey, BC, Canada).

## Quantitative Real Time (qRT)-PCR

Total RNA was prepared from testes using the PureLink Kit (Ambion). qRT-PCR was performed using SYBR Green Master Mix with gene-specific primer sets in a One-step RT-PCR reaction (Applied Biosystems) on an equal amount of total RNA from testes of each genotype and analyzed using ABI 7500 system sequence detection software (Applied Biosystems, Carlsbad, California). Arcturus Pico-Pure RNA kit from Applied Biosystems (Waltham, MA) was used to isolated RNA from the FAC-sorted cells.

## mRNA sequencing and analysis

An equal amount (4 µg) of total RNA from testes of 4 month-old mice was used for sequencing using the TruSeq RNA Sample Prep Kit v2. The libraries were amplified by 15 PCR cycles and quantitated using a Kapa Biosystems Quantification Kit (https://www.kapabiosystems.com/). The quality of each library was assessed on a Bioanalyzer Agilent DNA 1000 Chip. The libraries were then normalized to 2 nM and then pooled into one sample. The pooled samples were loaded onto 3 lanes of a V3 Flow Cell and then sequenced on the Illumina HiSeq 2000 (100 bp paired-end run). RNAseq data for all samples were subjected to quality-control check by NGSQCtoolkit v 2 (*Patel and Jain, 2012*), and samples with base qualities $\geq$ 30 over 70 nucleotides (100 bp reads) were used in the analysis. Orphaned reads after the quality-control step were used as single-ended reads in the downstream analysis. Read mapping was carried out using *TopHat v 2.0.7* (*56*) with supplied annotations at parameters (`-read-mismatches 2`, `-read-gap-length three` and `-read-edit-dist 3`) against the mouse genome (build-mm9). Bamtools v 1.0.2 (*57*) was used to calculate the mapping statistics. *HTSeq-count* script was used at default parameters (http://www-huber.embl.de/users/anders/HTSeq/doc/count.html) to count the reads mapped to known genes for both orphaned and paired-end reads, and final counts were merged prior to differential expression analysis. Expression differences between *Tg(Gdnf)* and *Tg(Gdnf);lu/lu* mice were determined by using *edgeR v 2.6.10* (*Robinson et al., 2010*) (*exactTest* function was used to determine the differences in expression). Genes with FDR $\leq$ 0.05 and Absolute $\log_2$ fold change $\geq$ 1 were considered differentially expressed.

## Regeneration of SSCs after busulfan treatment

A single dose of 10 mg/kg body weight of busulfan was injected intraperitoneally in WT C57BL/6J adult mice. Three mice were analyzed at each time point. Their testes were processed for whole-mount immunofluorescence. For a percentage of GFRA1 population, 300–500 cells were counts for each mouse. The EOMES+ and PLZF+ cells were counted over equal length of seminferous tubules, for each mouse. The data are presented as the number of cells ± SE per 1,000 Sertoli cells.

## Flow cytometry and single cell sequencing

Single cell suspensions from *Plzf* [+/+] and *Plzf* [lu/lu] testes carrying the *Eomes*-[tdTomato] knockin allele were prepared using a two-step collagenase and trypsin enzymatic digestion procedure. Cells were filtered first with an 40 um nylon membrane followed by a 25 um filter. DNase I was added to the cell suspension to prevent clumping. Cells were washed once in PBS and finally suspended at a concentration of $1-2 \times 10^6$ cells/ml, analyzed and sorted on Becton Dickinson FACSVantage cell sorter. Cells from littermate mice that were negative for *Eomes*-[tdTomato] were used to gate the positive staining cell population and DAPI was used to gate out the dead cells. Single cells were flow sorted into individual wells of a Bio-Rad hard shell 384 well plate. The plate was immediately transferred and stored in a −80C freezer. Custom designed Drop-Seq barcodes (see below) from Integrated DNA Technologies (IDT) were delivered into each well of each 384 well plate. All primers in one well shared the same unique cell barcode and billions of different unique molecular identifiers (UMIs). An Echo 525 liquid handler was used to dispense 1 ul of primers and reaction reagents into each well in the plate for the cell lysis and cDNA synthesis. Following cDNA synthesis, the contents of each well were collected and pooled into one tube using a Caliper SciClone Liquid Handler. After treatment with exonuclease I to remove unextended primers, the cDNA was PCR amplified for 13 cycles. The cDNA was fragmented and amplified for sequencing with the Nextera XT DNA sample prep kit (Illumina) using custom primers (see below) that enabled the specific amplification of only the 3′ ends. The libraries were purified, quantified, and sequenced on an Illumina NextSeq 500.

Barcode oligo primer: 5′-AAGCAGTGGTATCAACGCAGAGTACJJJJJJJJJJJJ NNNNNNNNTTTTTTTTTTTTTTTTTTTTTTTTTTTTTTTVN-3′
Custom primer sequence: 5′-AATGATACGGCGACCACCGAGATCTACACGCCTG TCCGCGGAAGCAGTGGTATCAACGCAGAGT*A*C-3′
Custom read 1 sequence: 5′-CGGAAGCAGTGGTATCAACGCAGAGTAC-3′

## Single cell sequencing analysis

Cells derived from the mutant and control samples were clustered separately. Genes with fewer than two counts in fewer than 2 cells were removed from the digital expression matrix and the expression was normalized by relative library size (computed as total counts per cell divided by median counts per cell) and log transformed. 100 genes with the largest variance/mean ratio and 11 genes used for cell type verification (*Gfra1*, *Eomes*, *Id4*, *T*, *Lin28a*, *Plzf*, *Bmi1*, *Pax7*, *Pdx1*, *Nanos2* and *Nanos3*) were used to subset the expression matrix for dimensionality reduction and clustering. Cellular expression profiles at these 110 genes were embedded into a 2-dimensional latent space using UMAP (McInnes L and Healy J. Uniform manifold approximation and projection for dimension reduction. ArXiv e-prints 2018, version 0.2.1), and clusters of cells were identified using hierarchical density-based spatial clustering (HDBSCAN, Campello Ricardo JGB et al., Hierarchical density estimates for data clustering, visualization, and outlier detection. ACM transactions on Knowledge Discovery from Data (TKDD) 2015; McInnes L, Healy J, and Astels S. Hierarchical density based clustering. Journal of open Sourse Software (JOSS) 2017, version 0.8.12). Cells assigned to cluster 0 are those which could not be confidently assigned to any cluster. Differentially expressed genes (DEGs) were identified using edgeR. Mutant and wildtype cells were treated as biological replicates for their respective condition. We find that edgeR offers a balance between type I and type II error rates when applied to single cell RNA-seq data relative to available scRNA-seq DEG analysis tools (*Dal Molin et al., 2017*).

## Acknowledgements

We are extremely grateful to M Havrda and M Israel for providing us with *Id4*[-/-] mice, to D Krawchuk for her help in manuscript preparation and to The Single Cell Biology Lab at The Jackson Lab for Genomic Medicine.

## Additional information

### Funding

| Funder | Grant reference number | Author |
|---|---|---|
| Eunice Kennedy Shriver National Institute of Child Health and Human Development | HD042454 | Robert E Braun |
| National Cancer Institute | CA34196 | Robert E Braun |

The funders had no role in study design, data collection and interpretation, or the decision to submit the work for publication.

### Author contributions

Manju Sharma, Conceptualization, Formal analysis, Investigation, Methodology, Writing—original draft, Writing—review and editing; Anuj Srivastava, William F Flynn, Data curation, Formal analysis; Heather E Fairfield, Investigation; David Bergstrom, Conceptualization, Investigation, Project administration; Robert E Braun, Conceptualization, Supervision, Funding acquisition, Writing—original draft, Project administration, Writing—review and editing

### Author ORCIDs

William F Flynn ⓘ https://orcid.org/0000-0001-6533-0340
Robert E Braun ⓘ https://orcid.org/0000-0003-3856-9465

### Ethics

Animal experimentation: All experimental procedures were approved by the Jackson Laboratory Institution Animal Care and Use Committee (ACUC) (protocol 07007) and were in accordance with accepted institutional and government policies.

### Decision letter and Author response

Decision letter https://doi.org/10.7554/eLife.43352.025
Author response https://doi.org/10.7554/eLife.43352.026

## Additional files

### Supplementary files

• Transparent reporting form
DOI: https://doi.org/10.7554/eLife.43352.019

### Data availability

Single cell RNA sequencing data have been deposited in GEO under accession code GSE116001. Total testis RNA sequencing data have been deposited in SRA under accession code PRJNA475219. All data generated or analyzed during this study are included in the manuscript and supporting files. Source data files have been provided for Figures 1, 2, 4, 5 and Figure 1-figure supplement 1.

The following datasets were generated:

| Author(s) | Year | Dataset title | Dataset URL | Database and Identifier |
|---|---|---|---|---|
| Sharma M, Srivastava A, Fairfield HE, Bergstrom D, Flynn WF, Braun RE | 2018 | Identification of EOMES-expressing spermatogonial stem cells and their regulation by PLZF | http://www.ncbi.nlm.nih.gov/geo/query/acc.cgi?acc=GSE116001 | NCBI Gene Expression Omnibus, GSE116001 |
| Sharma M, Srivastava A, Fairfield HE, Bergstrom D, Flynn WF, Braun RE | 2019 | Identification of EOMES-expressing spermatogonial stem cells and their regulation by PLZF | https://www.ncbi.nlm.nih.gov/bioproject/PRJNA475219 | Sequence Read Archive, PRJNA475219 |

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
