## [Decision Letter]

Thank you for submitting your article "Identification of slow-cycling germline stem cells and their regulation by PLZF" for consideration by *eLife*. Your article has been reviewed by three peer reviewers, and the evaluation has been overseen by a Reviewing Editor and Didier Stainier as the Senior Editor. The following individuals involved in review of your submission have agreed to reveal their identity: Shosei Yoshida (Reviewer #1); John Schimenti (Reviewer #3).

The reviewers have discussed the reviews with one another and the Reviewing Editor has drafted this decision to help you prepare a revised submission.

Summary:

In this report the authors take a careful and unbiased set of approaches to convincingly identify a sub-populaton of GFa1 cells with functional characteristics of SCC that label slowly and express the Tbox transcription factor Eomesodermin. Via scRNA-seq and other labelling experiments the data provided support the growing realization that the spermatagonial As population is heterogeneous in terms of gene expression and also plastic in terms of stemness states. This paper should make a significant contribution to the understanding of spermatagonial stem cells.

Essential revisions:

1) All three reviewers queried whether the data provided convincingly show that the *Eomes*+ cells do in fact represent slow-cycling cells. In light of the difficulty using existing reagents and experimental approaches to execute a "killer experiment" to definitively show this, the paper should be edited to remove this paragraph and reference to these data from the Abstract and remainder of the text. Leaving out the matter of long/slow cycling SSC does not make the manuscript less interesting.

2) If possible, and assuming the data is in hand, we recommend making a comparison between the expression levels of *Eomes* in ID4 bright and dim cells. This might hopefully add to the various findings from other groups regarding the identities of SSCs.

---

## [Author Response]

Essential revisions:1) All three reviewers queried whether the data provided convincingly show that the Eomes+ cells do in fact represent slow-cycling cells. In light of the difficulty using existing reagents and experimental approaches to execute a "killer experiment" to definitively show this, the paper should be edited to remove this paragraph and reference to these data from the Abstract and remainder of the text. Leaving out the matter of long/slow cycling SSC does not make the manuscript less interesting.

As requested, we have removed the data demonstrating that the EdU-labeling index significantly differs between EOMES+ cells and GFRA1+ cells. We have also removed text suggesting that EOMES+ cells are slow-cycling long-term SSCS. We have retained the data unequivocally demonstrating differential EdU-labeling of EOMES+ cells between *Plzf ^lu/lu^* and *Plzf ^+/+^* mice.

2) If possible, and assuming the data is in hand, we recommend making a comparison between the expression levels of Eomes in ID4 bright and dim cells. This might hopefully add to the various findings from other groups regarding the identities of SSCs.

Helsel et al., 2017, performed RNAseq on ID4 bright and dim cells. In Figure 3 of their paper, they did not report that *Eomes* was differentially expressed between the two groups. However, in response to the reviewers’ request, we reviewed the Helsel et al. source file (Database S1) and discovered that there is a 4.15 log2 fold (p = 0.00005) enrichment of *Eomes* in the ID4 bright population, suggesting a strong overlap in EOMES-expressing cells and the ID4 bright population. We have added this new information to the Discussion.